# Human Adenovirus Type 5 Infection Leads to Nuclear Envelope Destabilization and Membrane Permeability Independently of Adenovirus Death Protein

**DOI:** 10.3390/ijms222313034

**Published:** 2021-12-02

**Authors:** Søren Pfitzner, Jens B. Bosse, Helga Hofmann-Sieber, Felix Flomm, Rudolph Reimer, Thomas Dobner, Kay Grünewald, Linda E. Franken

**Affiliations:** 1Leibniz Institute for Experimental Virology (HPI), Martinistraße 52, 20251 Hamburg, Germany; soeren.pfitzner@leibniz-hpi.de (S.P.); jens.bosse@cssb-hamburg.de (J.B.B.); helgahofmann@gmx.de (H.H.-S.); felix.flomm@leibniz-hpi.de (F.F.); rudolph.reimer@leibniz-hpi.de (R.R.); thomas.dobner@leibniz-hpi.de (T.D.); 2Centre for Structural Systems Biology, Notkestraße 85, 22607 Hamburg, Germany; 3Cluster of Excellence RESIST (EXC 2155), Hannover Medical School, Carl-Neuberg-Straße 1, 30625 Hannover, Germany; 4Hannover Medical School, Institute of Virology, Carl-Neuberg-Straße 1, 30625 Hannover, Germany; 5Universität Hamburg, Institute for Biochemistry and Molecular Biology, Martin-Luther-King-Platz 6, 20146 Hamburg, Germany

**Keywords:** adenovirus death protein, electron cryotomography, HAdV5, nuclear egress, nuclear envelope, membrane damage, serial block-face SEM

## Abstract

The human adenovirus type 5 (HAdV5) infects epithelial cells of the upper and lower respiratory tract. The virus causes lysis of infected cells and thus enables spread of progeny virions to neighboring cells for the next round of infection. The mechanism of adenovirus virion egress across the nuclear barrier is not known. The human adenovirus death protein (ADP) facilitates the release of virions from infected cells and has been hypothesized to cause membrane damage. Here, we set out to answer whether ADP does indeed increase nuclear membrane damage. We analyzed the nuclear envelope morphology using a combination of fluorescence and state-of-the-art electron microscopy techniques, including serial block-face scanning electron microscopy and electron cryo-tomography of focused ion beam-milled cells. We report multiple destabilization phenotypes of the nuclear envelope in HAdV5 infection. These include reduction of lamin A/C at the nuclear envelope, large-scale membrane invaginations, alterations in double membrane separation distance and small-scale membrane protrusions. Additionally, we measured increased nuclear membrane permeability and detected nuclear envelope lesions under cryoconditions. Unexpectedly, and in contrast to previous hypotheses, ADP did not have an effect on lamin A/C reduction or nuclear permeability.

## 1. Introduction

Human adenovirus type 5 (HAdV5) belongs to the species C of mastadenoviruses and is a nonenveloped double-stranded DNA virus. So far, no exact mechanism has described how the mature adenovirus particles are released from infected cells. There are two processes present, the first to allow mature virions to cross the nuclear envelope and the second to exit the cell across the plasma membrane, both of which are possibly interlinked. In the case of adenoviruses, virions are released into the extracellular space after cellular lysis. Multiple cell-death pathways have been suggested to be the cause of ultimate host cell lysis. Although the process was initially described as apoptosis, a lack of membrane blebbing and DNA fragmentation hinted against apoptosis and another necroptosis-like cell death was suggested [1,2,3]. Viruses that generate progeny inside the nucleus employ several techniques to overcome the nuclear barrier, such as the formation of membrane pores [4], nuclear breakdown [5], export across nuclear pore complexes [6] or processes of envelopment/de-envelopment [7,8]. In HAdV infection, nuclear membrane damage has been proposed to be induced by the adenovirus death protein (ADP) preceding total cell lysis [1,9].

ADP of HAdV5 is a small 10.5 kDa viral protein. Its post-translational modifications have mainly been studied in its HAdV2 counterpart, which was shown to be N-glycosylated, O-glycosylated and palmitoylated [10,11]. The protein was described to initially localize to the endoplasmic reticulum (ER) and Golgi apparatus but ultimately is transported to the nuclear envelope. Therefore, it has been proposed to act on the nuclear envelope rather than the plasma membrane [10]. Deletion of ADP induced a phenotype of small plaques in adherent cell layers and extended survival of infected cells for multiple days. Overexpression of ADP was shown to increase virus spread and cell death [12,13].

Analysis of the spread of HAdV2 and HAdV5 in monolayer cell culture showed the virus to mainly spread by cell-free transmission [14,15]. Plaque formation was shown to be dependent on the lysis of infected cells and free diffusion of virus particles in the extracellular medium. Through this effect, comet-shaped plaques appear when the surface of monolayer cell culture is tilted [14,15]. Although ADP is not the sole determinant of lytic infection, it can increase cell-free spread. However, no direct evidence exists for the proposed membrane lytic properties of ADP [16]. While studies have reported membrane damage in classical electron microscopy of fixed samples, the nuclear envelope damage in HAdV infection has only been sparsely studied [17,18,19].

In this study, we have compared HAdV5 infection of HAdV5 strains with and without the presence of ADP in the viral genome. We probed lamin A/C localization at the nuclear envelope and measured the nuclear permeability during infection using a functional fluorescence microscopy assay. Furthermore, we analyzed infection-induced effects on nuclear membrane morphology using fluorescence microscopy, serial block-face scanning electron microscopy (SEM) and electron cryotomography. We found that infection leads to reduced lamin A/C levels and increased nuclear permeability, which coincided with membrane destabilization events and lesions in the nuclear envelope. Importantly however, we did not find a significant effect of ADP depletion on these processes.

## 2. Results

### 2.1. Expression of ADP Facilitates Release of Infectious Virus Particles but Is Unlikely to Result from ADP Pore Formation

The egress of HAdV5 particles has been linked to the protein ADP [1,9,12]. The commonly used laboratory strains of HAdV5 have a deletion in the E3 region since most of the E3 proteins are required for host cell immune evasion and are not necessary for cell culture systems. The wt HAdV5 ∆E3 strain is labeled HAdV5 ADP- in this work. For expression of ADP, the protein was reinserted in the E3 region by homologous recombineering in *E. coli* [20,21]. Virus particles were generated by transfection of the linearized viral genome including ADP into H1299 cells. The virus strain was passaged three times by infection of A549 cells with infection supernatant of the previous infection cycle. The resulting strain was labeled HAdV5 ADP+ (Figure 1a). To demonstrate the effect of ADP expression by this strain, virus release growth curves were measured. To this end, A549 cells were infected with HAdV5 ADP− and HAdV5 ADP+ strains and the supernatant medium was collected every 24 h from 0 hpi to 96 hpi. The titer of each supernatant was determined by FACS analysis (Figure 1b). 

At 24 hpi, no HAdV5 particles were detected in the supernatant for either virus strain. At 48 hpi, virus particles were released from cells infected with HAdV5 ADP+. From 48 hpi to 96 hpi, increasing numbers of infectious particles were found in the supernatant for HAdV5 ADP+. HAdV5 ADP− particles were only detected at 96 hpi at four-fold lower levels compared to HAdV5 ADP+. This time-course experiment demonstrated the increased release of virus particles upon ADP expression in infection. 

The mechanism of ADP-facilitated virion release is not known. Bioinformatic analyses of post-translational modifications (PTMs) and motifs indicate that ADP has a single transmembrane domain and two palmitoylation sites at residues Cys53 and Cys54 for further anchoring in the membrane (Appendix A). The luminal N-terminal side has predicted PTMs including N-glycosylation at residue Asn3 and O-glycosylation at residues Thr2, 4, 5, 11, 14, 16 and 17. The luminal N-terminus is predicted to be intrinsically disordered, and a basic, proline-rich region in the cytoplasmic domain could act as a nuclear localization signal. Amphipathic helices in viral proteins have been shown to lead to membrane pore formation aiding the release of viral particles [4,22,23]. The position and properties of the amino acid residues within the ADP transmembrane domain were analyzed but did not show properties of amphipathic helices that could oligomerize to form pores (Appendix A). Prediction of the ADP structure by the new deep-learning-based program AlphaFold2 [24] indeed provided a structure that has a hydrophobic domain, but lacks amphipathic properties (Appendix A). Structural comparison between HAdV5-ADP and HAdV2-ADP shows that with exception of the Nt and Ct tail-domains, the structure, including the transmembrane helix and the proline-rich region, is very conserved (Appendix A). While the Alphafold2 predictions are not confident, especially for the predicted disordered domains of the proteins, they hint towards potential structural differences. Aligning the Alphafold2 predictions based on the two transmembrane helices shows that the helix of HAdV2 ADP is predicted to be more curved in the luminal domain compared to ADP of HAdV5 (Appendix A).

ADP from HAdV2 was localized in the nuclear envelope where antibody staining resulted in a ring stain around the nucleus, as well as bright foci in the Golgi and ER, which possibly connects it to nuclear envelope destabilization [10,25]. To investigate ADP localization in HAdV5 and follow ADP during infection, an ADP–HA fusion was generated in the context of the HAdV5 genome by homologous recombineering in *E. coli* [20], yielding the virus strain HAdV5 ADP–HA. Mutant virus particles were generated by transfection of the linearized viral genome into H1299 cells. After transfection, the cells were analyzed for cytopathic effects. The culture supernatant was used to infect new A549 cells, and progeny virions were produced by three rounds of infection. Since no commercial antibodies against ADP exist and antibodies from former studies [10,25,26] were no longer available or functional against HAdV5 ADP, the HA tag allowed us to explore HAdV5 ADP localization in infection by immunofluorescence microscopy at 48 hpi (Appendix A). Unexpectedly, ADP–HA was not detected in the nuclear envelope, but was found to be exclusively localized in cytoplasmic spots that resembled vesicles. These vesicles appeared to be associated with the perinuclear region (i.e., ER or Golgi apparatus), which points towards a localization that is distinctly different from what has been described for HAdV2 [10]. 

### 2.2. Infection Reduces Lamin A/C Levels at the Nuclear Envelope Independently of ADP Expression

Since fluorescent labeling of ADP did not reveal specific regions of the nuclear envelope to be enriched with ADP protein, the overall nuclear envelope morphology of infected cells was compared in HAdV5 ADP− and HAdV5 ADP+ infection at 48 hpi. We previously described an A549 cell line which expresses a lamin A/C nanobody fused to mTagGFP [27]. This cell line was used to analyze the morphology of the nuclear envelope (Figure 2a). Additionally, the lamin A/C fluorescence intensity levels at the nuclear envelope were quantified and compared between noninfected cells and infection with either HAdV5 ADP− or HAdV5 ADP+ strains at 48 hpi (Figure 2b). 

Noninfected A549 cells have a clear ring staining of lamin A/C around the nucleus. In infected cells, the lamin A/C signal was strongly reduced and the nuclei lost their round shape. Infected cells were detected by the presence of the honeycomb-like dsDNA conformation of the LVAC in the Hoechst 33,342 channel. The LVAC forms late in HAdV5 infection and is a good indicator of an advanced stage of the infection [27]. No clear difference in lamin A/C reduction or nuclear shape could be detected between HAdV5 infection with and without ADP. To quantify the lamin A/C signals, the nuclear lamina was manually segmented, and the average signal was measured across the entire nuclear envelope. No significant difference could be detected between the HAdV5 strains. All infected cells showed significantly reduced lamin A/C signal compared to noninfected cells. These results indicated an infection-dependent reduction of lamin A/C, which is not modulated by ADP expression.

### 2.3. Serial Block-Face SEM and Electron Cryotomography Reveal Details of HAdV5-Induced Nuclear Membrane Deformation

In addition to the reduction in lamin A/C fluorescence levels, the nuclear envelope of infected cells showed a loss of structural integrity. In noninfected cells, the nuclear membrane had an oval shape, whereas the nuclear envelope of infected cells developed into a kidney shape and showed membrane invaginations. To further characterize the membrane deformations caused by HAdV5 infection, these nuclear membrane invaginations were analyzed using serial block-face SEM. A noninfected cell and a HAdV5 ADP− and ADP+-infected cell each were recorded as sequential z-slices to reconstruct the 3D volume of the entire cell. To identify appropriately infected cells in a correlative approach, a HAdV5 strain expressing mCherry-labeled pV was used [27]. By tracing and segmenting the nuclear envelopes of measured cells, the global nuclear shape was analyzed. Since no difference in large-scale nuclear envelope morphology between ADP+ and ADP− infected cells was identified (data not shown), the ADP+ infected cell was analyzed representatively for infection to compare the global nuclear shape with the noninfected cell (Figure 3a).

The noninfected cell had an oval-shaped nucleus and contained a nucleoplasmic reticulum as a small channel, spanning one side of the nucleus to the other (Figure 3a images (i) and (ii)). Overall, the nuclear envelope showed no major deformations (Figure 3a image (iii)). In contrast, the infected cell showed fragmentations of the nucleus into multiple parts (Figure 3a images (iv) and (v)). Overall, the nucleus of the infected cell had a kidney-shaped morphology. In particular, the part of the nuclear envelope which was facing the center of the cell showed strong membrane invaginations and budding events of multiple small vesicles containing nucleoplasm (Figure 3a image (vi)).

To achieve higher resolution of the nuclear double membrane and to analyze the phenotype of the nuclear envelope in more detail, we used electron cryotomography of focused ion-beam (FIB) milled cells. To this end, we infected A549 cells with HAdV5 ADP+ expressing mCherry-labeled pIX [27], plunge-froze them and used the pIX-mCherry fluorescence signal to select appropriately infected cells for production of thin lamellae of the nuclear region by milling. We then recorded regions of interest by acquiring transmission electron cryomicroscopy tilt series and reconstructed them as 3D tomograms (Figure 3b). 

Multiple small-scale membrane instabilities were detected in infected cells. One observable phenotype was the occurrence of sites at which both membranes of the nuclear envelope came into close contact. These contacts either appeared in small areas (Figure 3b image (i)) or over long membrane stretches (Figure 3b image (iv)). Another observable phenotype was the protrusion of the outer membrane into the cytoplasm away from the inner membrane (Figure 3b images (ii) and (v)). Additionally, simultaneous protrusions of both membranes were observed (Figure 3b images (iii) and (vi)). All these phenotypes indicate that the nuclear membrane lost stability during infection, a process which might be responsible for subsequent nuclear membrane rupture.

### 2.4. Adenovirus Infection Leads to Damage of the Nuclear Envelope and an Increased Nuclear Permeability Independent of ADP

In the next step, additional serial block-face SEM data was recorded to look into the potential role of ADP in nuclear envelope disruptions. This method records the entire volume of the infected cell, thus it increases the chances of finding regions of nuclear envelope damage. A549 cells were infected with HAdV5 pIX-mCherry ADP− and HAdV5 pIX-mCherry ADP+. Appropriately infected cells were detected via their pIX-mCherry fluorescence signal at 42 hpi and prepared for serial block-face SEM (Figure 4).

Infection with virus lacking or expressing ADP resulted in sites of nuclear envelope damage, which appeared as gaps in the nuclear double membrane (Figure 4a). In contrast, the nuclear envelope of a noninfected cell did not show any signs of membrane damage. By analyzing sequential z-slices of the SEM 3D volume, the sites of nuclear envelope damage were shown to extend over multiple z-slices rather than being single damage events in single z-planes (Appendix A). While more of these sites were detected in infection with HAdV5 ADP+ than HAdV5 ADP− (data not shown), the number of analyzed cells does not allow for a meaningful quantification. However, since no damage was detected in noninfected cells, it is unlikely that they arise from SEM-preparation-induced artifacts.

Since the throughput of most EM methods is limited and in order to link the structural and functional data, we measured the direct influence of ADP expression on the permeability of the nuclear envelope by a fluorescence microscopy-based assay. The assay was based on the nuclear influx of 70 kDa dextran molecules labeled with the fluorescent dye fluorescein isothiocyanate (FITC). Dextran-FITC molecules are not able to pass intact nuclear pore complexes (NPCs) and can thus only reach the nucleoplasm through enlarged nuclear pores or membrane lesions. The dye is not cell-permeable, which is why cells were gently pretreated with digitonin to selectively permeabilize the plasma but not the nuclear membrane. As a positive control, cells were incubated in triton X-100 to also permeabilize the nuclear membrane. The A549 cells were infected with HAdV5 pIX-mCherry ADP− and HAdV pIX-mCherry ADP+, semipermeabilized, incubated in dextran-FITC and then analyzed by spinning-disk confocal fluorescence microscopy (Figure 5a).

Noninfected cells showed dextran-FITC fluorescence in the space between cells as well as the cytoplasm. Dextran-FITC signal was weaker inside the cytoplasm compared to the extracellular space, and the dye was excluded from intracellular vesicles. The dextran-FITC fluorescence inside the nucleus was very weak. Hoechst 33,342 clearly stained cellular chromatin inside the nucleus. The positive control showed stronger dsDNA signal due to the increased permeation of the dye, which marks the nuclear area. The fluorescence of dextran-FITC appeared to be higher compared to the negative, noninfected control. In infection, Hoechst 33,342 stained the marginalized dsDNA as well as viral DNA that accumulated in the LVAC. Dextran-FITC signal was detected inside the nucleus, however, it did not show an even distribution. Instead, subnuclear areas outside of the LVAC showed high dextran-FITC fluorescence, while the probe was excluded from the LVAC. To quantitatively analyze these results, we used image segmentation. The nuclear area was selected by use of the dsDNA stain Hoechst 33,342, and the whole-cell area was selected based on the cellular outline that was visible in the dextran-FITC channel (Figure 5b). Subsequent quantification of the dextran-FITC ratio between cytoplasm and nucleus showed the average signal ratio to be 1.25 for noninfected cells, 1.11 for the positive triton X-100 control, 1.03 for infection without ADP and 1.03 for infection with ADP (Figure 5c). In this assay, the nuclear permeation of the dye was significantly increased by infection. No statistically significant difference in nuclear permeability between infection with and without ADP was detected. 

### 2.5. Nuclear Envelope Lesions Are also Detectable under Cryocondition in Electron Cryotomograms of FIB-Milled Infected Cells

To confirm the presence of nuclear envelope lesions under more native cryoconditions compared to serial block-face SEM, the nuclear envelopes of HAdV5 ADP+-infected cells were analyzed for membrane damage by electron cryotomography. For better visualization, regions of interest of the nuclear envelope were segmented (Figure 6).

Three different kinds of membrane openings were observed: (i) normal nuclear pore complexes (NPCs) (Figure 6a), (ii) nuclear holes of a larger size than NPCs (Figure 6b) and (iii) membrane damage, similar to the events previously detected by serial block-face SEM (Figure 6c). The tomogram of a site of nuclear membrane damage included a cytoplasmic region flanked by two nucleoplasmic regions caused by nuclear envelope infolding. Within the volume of the tomogram, a high number of virions was detected in the nucleoplasm, but three particles were also detected in the cytoplasm. These particles were positioned close to an opening in the nuclear envelope, which is larger than the typical NPC diameter and opened up further to the top of the tomogram. These cryotomograms demonstrate that the nuclear damage as observed with serial block-face SEM is not an artefact of sample preparation, as it can also be visualized in native cryoconditions. These lesions likely explain the observed increased permeability of the nuclear envelope (Figure 5).

## 3. Discussion

The final steps of the HAdV lifecycle are the release of mature virions from the nucleus of infected cells into the cytoplasm and the subsequent spread to neighboring cells across the plasma membrane. This mechanism has been broadly described as lysis of the cell and release of cellular content into the surrounding extracellular space [15]. However, the exact egress mechanism is still elusive and the distinction with respect to the two important barriers that need to be overcome, i.e., nuclear and cellular egress, is rarely made in the context of adenovirus infection.

ADP has been shown to enhance viral spread and to enlarge plaque size upon increased protein expression [1,12]. We have shown that our ADP+ virus strain displayed those characteristics when comparing its virus titer growth curve to the ADP− strain (Figure 1b). ADP of HAdV2 was previously shown to localize to the nuclear membrane, but was also found in the ER and Golgi of A549 cells [10,29]. To study the localization of HAdV5 ADP, we labeled ADP with an HA tag. In contrast to previous reports, we observed that the HA-tagged protein was almost exclusively localized to the perinuclear region (Appendix A). These differences could either be a result of HAdV5 ADP naturally having a different localization to HAdV2 ADP or the addition of tags causing protein localization artifacts. This should be further tested by immunostaining against ADP. Bioinformatics comparison based on their amino acid composition indicates a fully conserved transmembrane helix, and a highly conserved central region, but much larger differences in the N-terminal and C-terminal domains that could potentially have consequences for function and localization (Appendix A). Interestingly, Georgi et al. proposed that ADP function might depend on the protein being localized and palymitoylated in Golgi vesicles rather than its presence in the nuclear membrane [29]. This hypothesis was derived from the fact that after addition of the inhibitor nelfinavir, which blocks ADP function, the protein still localized to the nuclear envelope but not cytoplasmic vesicles. In our hands, ADP-HA was still functional at releasing virus, even though we did not observe the protein in the nuclear envelope. Similarly, a mutant of HAdV2 ADP+ lacking a region of the C-terminal tail was also shown to lose nuclear envelope localization but still caused cell lysis [25]. In our experiments, we could neither confirm strong localization of ADP to the nuclear envelope, nor could we find a relationship between nuclear permeability/nuclear membrane damage and the presence of ADP, which is in line with its activity being related to cell lysis, rather than nuclear egress. 

Many viruses express proteins that possess membrane-modulating properties. In nonenveloped viruses, such proteins are involved in forming pores or disrupting the plasma or endosomal membrane. Examples are poliovirus VP1 and VP4, rotavirus VP4, reovirus penetration protein γ1, polyomavirus VP2 and VP3 or HAdV pVI [23,30,31,32,33]. These proteins are all required for overcoming the physical membrane barrier of the host cell during entry of the virus. However, much less is known about proteins modulating membranes during nuclear or cellular egress. While cell lysis as part of a cell-death pathway is often suggested to be the main driver for virion release, viral proteins are likely to control this process to ensure sufficient progeny-virion production before host cell death. In polyomaviruses, two proteins VP4 and agnoprotein have been reported to aid virus nuclear and cellular egress. Both proteins act as viroporins and can permeabilize membranes to allow for the passage of large molecules [34,35,36]. Agnoprotein has been shown to include an amphipathic helix which allows oligomerization within the membrane leaflet to produce a hydrophilic pore [37]. It is conceivable that HAdVs possess a similar mechanism for membrane pore formation. However, analysis of the transmembrane helix of HadV5 and HadV2 ADP showed that it does not form an amphipathic helix (Appendix A). More importantly, our analysis of the permeability of the nuclear membrane indicated an increased nuclear permeability, but this process was independent of ADP presence (Figure 5). Increased membrane permeability during HAdV2 infection also occurs during virus entry [38]. HAdV2 capsid disassembly at the NPC has been reported to increase nuclear pore permeability. Interestingly, the authors described a time-dependence of this effect in a way that dextran-FITC only permeated very early after infection of cells, whereas the dye could not permeate the nucleus at 4 hpi. It is therefore unlikely that the described effect is identical to our observation of increased nuclear permeability at 48 hpi.

The increased nuclear permeability at 48 hpi could be explained by the appearance of nuclear envelope lesions, which we observed in this study by two different state-of-the-art electron microscopy techniques, serial block-face SEM and electron cryotomography (Figure 4 and Figure 6c). By combining these techniques, potential artifacts that might occur during cell preparation for either method could be ruled out as a cause for membrane damage. Particularly, the membrane damage can be ascribed to the effect of HAdV5 infection since no membrane damage was detected in noninfected cells by block-face SEM analysis. Similarly, nuclear membrane damage in adenovirus infection has been described for mouse adenovirus infection of mouse adrenal glands. The authors showed viral progeny in the cytoplasm coinciding with large breaks in the nuclear envelope [18]. Additionally, nuclear envelope breaks were reported for HAdV2 and HAdV5 infection of HeLa cells [17,19]. While we did not observe virions directly within a gap of the nuclear envelope during the process of traversing, virus particles were observed in the cytoplasm close to nuclear envelope lesions (Figure 3b and Figure 6c, Appendix A), which hints at the particle release from the nucleus at these sites. 

Next to nuclear envelope damage, increased nuclear permeability might result from dilated nuclear pores. In our electron cryotomography analysis we observed nuclear envelope openings of varying size (Figure 6a,b). Nuclear pore dilation also occurs in influenza virus infection to enable vRNP nuclear egress [39]. It is plausible that HAdV5 infection increases nuclear pore diameter and thus results in the observed increase in nuclear membrane permeability (Figure 5). While we did not observe a difference in permeability upon ADP expression, ADP could contribute to further increased nuclear pore diameters or greater damage at sites of virus-induced nuclear envelope breakdown. This would not be detected in our permeability assay, because the dye is smaller than a viral capsid, but could in theory explain why electron microscopy studies have only reported virions outside of the nucleus in HAdV infection when ADP was expressed [1,9,19]. However, our ADP–HA localization experiment points towards a more cellular role for ADP (Appendix A), and the lack of viral particles in the cytoplasm at comparable time points that have been observed previously could rather reflect the general delay in viral growth and release in ADP-deletion strains (Figure 1b).

Importantly, our electron microscopy analysis did not find evidence for alternative egress events such as nuclear membrane punctures by large viral protein crystals, or, as has been shown for HSV-1, a process of envelopment/de-envelopment [7,8].

As part of a more global analysis of the nuclear membrane integrity by serial block-face SEM, we observed large-scale membrane instabilities (Figure 3a). The destabilization of the nuclear envelope by HAdV infection could be a prerequisite for nuclear rupture. When analyzing nuclear lamin A/C, a clear reduction in signal at the nuclear envelope could be detected (Figure 2). Again, this effect was observed independently of ADP expression. Lamin A/C has been shown to be important for nuclear stability since lamin mutations cause nuclear deformation as well as local nuclear envelope rupture [40,41,42]. Similarly, a reduction in nuclear lamins by human cytomegalovirus infection was shown to induce nuclear membrane infoldings [43]. HAdV5 might cause lamin A/C degradation as part of an infection-induced cell-death process. Lamin A/C has been described to be cleaved by caspases during apoptosis [44]. Alternatively, lamin A/C cleavage could be caused by a viral protein such as adenoviral protease (AVP), which has been shown to cleave cytokeratin 18 to disrupt the cytokeratin network of infected cells. This process has been hypothesized to lead to cellular destabilization and more efficient virus release [45]. Indeed, our electron microscopy analysis of the nuclear membrane in HAdV5 infection showed frequent nuclear membrane invaginations, protrusions, variations in double-membrane separation distance, as well as breaks in the nuclear envelope (Figure 3, Figure 4 and Figure 6, Appendix A). Although there seems to be no role for ADP in the reduction of lamin A/C, the destabilization of the lamin network may be a prerequisite to the process of nuclear egress, and is in itself worthy of further studies. At the moment it is not known whether lamin A/C is degraded, translocated or whether nuclear swelling causes it to lose its integrity. Similarly, it is not known whether phosphorylation plays a role in this, and which viral proteins drive this process. Furthermore, the localization and fate of lamin B as well as the fates of several nuclear envelope proteins, e.g., Emerin or Lap2, could be relevant targets for future studies to further our understanding of the extent and mechanism of nuclear destabilization that is required for nuclear egress.

Our observations about lamin A/C localization, SEM and electron cryotomography imaging of the nuclear envelope of HAdV5-infected cells and observations of virus particles outside of the nucleus can be summarized in a model about adenovirus nuclear egress (Figure 7). Upon infection, the lamin A/C stabilizing network of the nucleus is degraded or redistributed such that the nuclear envelope loses stability. Consequently, the nuclear envelope develops large-scale invaginations, and small-scale double-membrane perturbances and protrusions. In addition to these nuclear envelope instabilities, sites of nuclear damage occur, which are visible as lesions in SEM and electron cryotomography imaging. Virus particles are not associated with these lesions in all cases, indicating that the lesions do not occur due to direct particle interactions. Single virus particles can escape via these lesions and are detected in the cytoplasm. Even later in infection, the nuclear membrane ruptures more extensively leading to release of larger amounts of nuclear material, including virus particles, into the cytoplasm (Appendix A). The virus particles are ultimately released into the surrounding extracellular space when cells lyse due to cell death.

In summary, we show that during HAdV5 infection the nuclear envelope and its supporting lamin network are heavily altered prior to viral release. We applied a combination of light microscopy and electron microscopy techniques and observed reduced lamin A/C levels at the nuclear envelope, increased nuclear membrane permeability and multiple nuclear membrane destabilization events on small and large scales. Notably, and in contrast to our expectations when starting to analyze HAdV5 infection, we did not observe a dependency of these events on the presence of ADP. However, since ADP expression does increase the virus release from infected cells, a different role for ADP must exist. 

## 4. Materials and Methods

### 4.1. Mammalian Cell Line Culture

Mammalian cell lines A549 (ATCC CCL-185), A549 lamin A/C mTagGFP [27] and H1299 (ATCC CRL-5803) were cultured in Dulbecco’s Modified Eagle’s Medium (DMEM) (Gibco DMEM, high glucose, pyruvate, Thermo Scientific™, Thermo Fisher Scientific, Waltham, MA, USA) with 1% (*v*/*v*) penicillin/streptomycin (P/S) solution (final 1000 U/mL penicillin and 1 mg/mL streptomycin, Pan-Biotech, Aidenbach, Germany) and 10% (*v*/*v*) fetal calf serum (FCS) (FCS Superior, Merck KGaA, Darmstadt, Germany). All cells were grown at 37 °C and 5% (*v*/*v*) CO_2_. 

### 4.2. Construction of Virus Strains and Expression Plasmids

The adenoviral mutants were derived from the adenovirus strain H5pg4100 [46], lacking 1863 bp (nt 28602-30465) of the E3 region which is termed HAdV5 ADP− in our study. The HAdV5 genome was modified and propagated in *Escherichia coli* within the low copy number plasmid p15A. Adenoviral variants carrying mCherry-labeled pIX and pV were characterized in our previous study [27]. They were generated via homologous recombination in a ccdB counterselection system [20]. ADP was inserted into the E3 region by homologous recombination to yield HAdV5 ADP+ constructs. Similarly, an HA tag was inserted at the C-terminus of ADP by homologous recombination (Table 1). 

### 4.3. Virus Production, Titration, Infection and Growth Curve

New recombinant virus particles were produced in H1299 cells. A 10 cm culture dish of 90% confluent cells was transfected with lipofectamine 2000 and 1 µg linearized adenoviral genome. 7 days after transfection, upon visible cytopathic effects, virus particles were isolated by three freeze–thaw cycles. Cell debris was removed by centrifugation at 3400× *g* for 15 min and the supernatant was used to start a new round of infection for further passaging of the recombinant virus. After five passages, the titer of the supernatant was determined via flow cytometry. A549 cells were seeded in 12-well plates and infected with a serial dilution of stock solution. After 24 h, the cells were trypsinized, washed in PBS and fixed in 4% (*v*/*v*) PFA solution. The cells were resuspended in FACS buffer (1% (*v*/*v*) FCS in PBS) and analyzed on an LSR Fortessa cell analyzer (BD Biosciences). The percentage of intact, individual fluorescent cells was used to determine the fluorescence-forming units (ffu) per µL of virus stock solution.

HAdV5 infection was performed at a multiplicity of infection (MOI) of 1 ffu/cell. For infection, the cells together with virus stock were incubated in DMEM (FCS and P/S free) for 1 h. Subsequently, DMEM with 1% (*v*/*v*) P/S and 10% (*v*/*v*) FCS was added in equal amounts, leading to a final supplement concentration of 0.5% (*v*/*v*) P/S and 5% (*v*/*v*) FCS.

HAdV5 virus release growth curves were performed in A549 cells. Cells were infected with an MOI of one in triplicate. The supernatant medium was carefully collected at 0, 24, 48, 72 and 96 hpi and stored at −80 °C. Next, 20 µL of supernatant was used to infect A549 cells. After 24 h, the cells were trypsinized, washed in PBS and fixed in 100% (*v*/*v*) methanol. The cells were blocked in FACS buffer (1% (*v*/*v*) FCS in PBS) and stained for DNA-binding protein using primary mouse B6-8 anti-DBP antibody [47]. Secondary antibody used was anti-mouse Alexa Fluor 488 (Invitrogen™, Thermo Fisher Scientific, Waltham, MA, USA). The cells were resuspended in FACS buffer and analyzed on an LSR Fortessa cell analyzer (BD Biosciences, San Diego, CA, USA). The percentage of intact, individual fluorescent cells was used to determine the fluorescence-forming units per µL of infection supernatant. 

### 4.4. Live-Cell Fluorescence Microscopy

A549 or H1299 cells were plated on culture dishes (µ-Dish 35 mm, glass bottom, ibidi) previously coated with 1% (*v*/*v*) fibronectin (Sigma Aldrich, Merck KGaA, Darmstadt, Germany) for 40 min. The cells were either transfected by addition of 1:10 (*v*/*v*) DNA:polyethylenimine for 5 h or infected with varying HAdV5 strains and prepared at 24 hpi or 48 hpi for imaging of proteins of interest. Images were recorded on an inverted confocal spinning-disk microscope (Nikon Eclipse Ti-2 stand; Yokogawa CSU-W1 spinning disk; 2× Andor888 EM-CCD camera; Nikon 100× oil-immersion numerical aperture (NA) 1.49 objective, Nikon Corporation, Tokyo, Japan) equipped with a heating chamber at 37 °C and 5% (*v*/*v*) CO_2_. Images were recorded using the Nikon NIS-Elements software. Further image processing was performed in Fiji [48] or Nikon NIS Elements software.

### 4.5. Immunofluorescence Microscopy

A549 cells were plated on culture dishes (µ-Dish 35 mm, glass bottom, ibidi) previously coated with 1% (*v*/*v*) fibronectin (Sigma Aldrich) for 40 min, infected with varying HAdV5 strains and prepared at 24 hpi or 48 hpi for analysis of localization of proteins of interest. The cells were washed in PBS, fixed with 4% (*v*/*v*) paraformaldehyde (PFA) in PBS for 20 min and washed with PBS again. Afterwards, the cells were quenched with 25 mM NH4Cl in H_2_O for 10 min, washed in PBS and permeabilized with 0.5% (*v*/*v*) Triton X-100 for 10 min. After a PBS wash, cells were blocked with 1× TBS-BG solution (20 mM Tris/HCL pH 7.6, 137 mM NaCl, 3 mM KCl, 1.5 mM MgCl2, 0.05% (*v*/*v*) Tween 20, 5 mg/mL bovine serum albumin, 5 mg/mL glycine). After a PBS wash, the cells were stained for 1 h with primary antibodies, washed with PBS and stained for 30 min with secondary antibodies and Hoechst 33342 (0.05% (*v*/*v*). Primary antibodies used were anti-HA (Roche, Merck KGaA, Darmstadt, Germany). Secondary antibodies used were anti-rat Alexa Fluor 488 (Invitrogen). Images were recorded on an inverted confocal scanning laser microscope (Nikon A1R HD25 equipped with a Nikon 60x oil-immersion NA 1.40 objective) using the Nikon NIS-Elements software. Further image processing was performed in Fiji [48].

### 4.6. Nuclear Permeability Assay

5 × 10^4^ A549 cells were plated on culture dishes (µ-Dish 35 mm, glass bottom, ibidi), which were coated with 1% (*v*/*v*) fibronectin (Sigma Aldrich) for 40 min, infected with varying HAdV5 strains and prepared at 48 hpi for analysis of the nuclear membrane permeabilization. Cells were first transferred onto ice and washed three times with ice-cold PBS. Afterwards the cells were incubated in ice-cold permeabilization buffer (PB) (2 mM HEPES (pH 7.5), 11 mM KOAc, 0.5 mM Mg (OAc)_2_, 50 µM EGTA, 250 mM sucrose) containing 24 mg/mL digitonin for 10 min. After digitonin incubation, the cells were incubated in ice-cold PB for 1 min, followed by 5 min and 10 min steps in PB, and finally a 10 min incubation step in transfer buffer (TRB) (2 mM HEPES (pH 7.5), 11 mM KOAc, 0.2 mM Mg (OAc)_2_, 0.5 mM NaOAc, 50 µM EGTA, 250 mM sucrose). The cells were then covered with a dextran solution (0.6 mg/mL 70 kDa dextran-FITC and 1:2000 Hoechst 33342 in TRB) and incubated for 10 min at room temperature before imaging. Images were recorded on an inverted confocal spinning-disk microscope (Nikon Eclipse Ti-2 stand; Yokogawa CSU-W1 spinning disk; 2× Andor888 EM-CCD camera; Nikon 100x oil-immersion NA 1.49 objective) equipped with a heating chamber at 37 °C and 5% (*v*/*v*) CO_2_. Images were recorded and processed using the Nikon NIS-Elements software. 

### 4.7. Serial Block-Face Scanning Electron Microscopy

A culture dish (μ-Dish 35 mm, high Grid-500, ibidi) was seeded with 1 × 10^5^ A549 cells and infected with varying HAdV5 strains. The cells were fixed at varying time points by the addition of 2% (*v*/*v*) PFA and 2.5% (*v*/*v*) glutaraldehyde (GA) in PBS for 5 min at room temperature. The cells were further fixed by addition of 2% (*v*/*v*) PFA and 2.5% (*v*/*v*) GA in PBS for 55 min 4 °C and washed with PBS for 1 min at room temperature once. The cells were postfixed in ice-cold 2% (*w*/*v*) OsO_4_ and 2.5% (*v*/*v*) GA in PBS for 30 min at 4 °C. After 5 washing steps in ice-cold PBS, cells were stained based on the osmium–thiocarbohydrazide–osmium (OTO) contrasting method [49]. Each contrasting step was followed by 10 washing steps in H_2_O for 1 min at room temperature. First, cells were contrasted with 2% (*w*/*v*) OsO_4_, 1.5% (*w*/*v*) potassium ferrocyanide and 2 mM CaCl_2_ in H_2_O for 1 h at room temperature. Then, cells were incubated in 0.5% (*w*/*v*) thiocarbohydrazide for 10 min at room temperature. Afterwards, cells were contrasted again in 2% (*w*/*v*) OsO_4_ for 20 min at room temperature. Further staining was achieved by incubation in 1% (*w*/*v*) gallic acid for 10 min at room temperature, 2% (*w*/*v*) uranyl acetate overnight at 4 °C and Walton’s lead aspartate solution (30 mM aspartic acid, 20 mM PbNO_3_) for 10 min at 60 °C. The cells were dehydrated according to the progressive lowering of temperature (PLT) principle [50]. First, cells were incubated in 30% (*v*/*v*) EtOH for 30 min at 0 °C. Then, the EtOH concentration was increased to 50% (*v*/*v*) while lowering the temperature to −20 °C for 30 min. Afterwards, the cells were incubated in 70% (*v*/*v*) EtOH for 30 min at 20 °C. The temperature was further lowered to −35 °C and cells were incubated in 100% (*v*/*v*) EtOH for 20 min. This step was repeated with fresh 100% (*v*/*v*) EtOH for 20 min at −35 °C. Finally, the cells were gradually embedded in epoxy resin. Cells were rocked in 1:1 (*v*/*v*) EtOH/epoxy resin for 30 min at 4 °C, followed by incubation in 1:1 (*v*/*v*) EtOH/epoxy resin for 1 h at room temperature and 100% (*v*/*v*) epoxy resin overnight at room temperature. The next day, cells were incubated in polymerization medium (3% (*v*/*w*) silver particles, 5% (*w*/*v*) Ketjenblack in epoxy resin) for 6 h at room temperature [51]. The medium was removed to a level so that only a thin layer was covering the cells and polymerization was carried out for 48 h at 60 °C. Polymerized cells of interest were cut out of the culture dish with a heated scalpel and mounted onto an epoxy resin stub using epoxy glue. Rough trimming was performed using a high-speed milling system to generate a flat-top pyramid with 4 sides at 45° angles. 160 μm of the ibidi polymer culture dish layer on the face of the flat-top pyramid was removed by serial sectioning on an ultramicrotome. The area of the pyramid block face was further reduced to about 0.25 mm^2^ by removing the excess material with a scalpel. The top of the pyramid was then cut away at a depth of about 1 mm and transferred to an aluminum stub. The sample block was attached to the stub by surrounding the sides and bottom of the block by conductive silver epoxy resin. The silver epoxy was polymerized overnight at 60 °C and the sides of the pyramid were fine-trimmed to retain a thin silver epoxy layer covering the sample. As a final step to increase conductivity of the sample surface, it was sputter-coated with an 8 nm layer of gold. The sample was then transferred to a serial block-face scanning electron microscope (JEOL Ltd., Tokyo, Japan) equipped with a 3view stage (Gatan, Inc., Pleasanton, CA, USA) operating at 3 kV. A biasing charge of +600 V was applied to the stage to reduce charging effects of the sample during prolonged image acquisition. The sample block face was sequentially trimmed by 50 nm, followed by acquisition of scanning electron microscopy images of the newly exposed surface using DigitalMicrograph software (Gatan, Inc.). The same software was used to align SEM image stacks for generation of 3D volumes. Further image processing was performed in Fiji [48]. Manual image segmentation was performed in IMOD [28]. 

### 4.8. Electron Cryotomography

Before seeding, holey carbon-coated gold finder grids (Quantifoil R2/1) were glow-discharged for 1.5 min. The grids were coated with 6 µL of 1% (*v*/*v*) fibronectin and left to dry under UV-light for sterilization for 1 h. The grids were remoisturized with PBS and placed in 2 × 9 chambered culture dishes (ibidi) and covered by 50 μL of DMEM (Gibco DMEM, high glucose, pyruvate, Thermo Scientific™, Thermo Fisher Scientific, Waltham, MA, USA) with 10% (*v*/*v*) FCS. A549 or A549 lamin A/C mTagGFP [27] cells were harvested from a 10 cm culture dish at 80% confluency by addition of stable trypsin replacement enzyme for 15 min. The reaction was quenched through addition of DMEM to a final ratio of 2:3 (*v*/*v*), the cells were pelleted at 670× *g* (Centrifuge 5920 R, Thermo Scientific) for 3 min and the supernatant was replaced with DMEM at equal volume. The cells were mixed with HAdV-5 pIX-mCherry ADP+ at an MOI of 1–2 and seeded on gold grids by addition of 5–15 μL of cell suspension to each well. At 36–48 hpi, the grids were imaged by live-cell fluorescence microscopy. Afterwards the cells were cryofixed by plunge freezing. For this, grids were removed from the growth chamber, wetted with 5 μL DMEM and transferred to a manual cryoplunger. Grids were blotted from the back for 8 s and plunged into a 37:63 (*v*/*v*) mixture of ethane/propane cooled to liquid nitrogen temperature. The cryofixed grids were clipped into Cryo-FIB Autogrids (Thermo Scientific) and loaded into an Aquilos Cryo-FIB microscope (Thermo Scientific) for FIB-milling. A correlation between live-cell fluorescence microscopy images was achieved by importing the images into MAPS software (Thermo Scientific) and overlaying them with an SEM tile scan. Appropriate sites for milling were selected based on the HAdV5 pIX-mCherry fluorescence signal. After an initial SEM scan, grids were sputter-coated with platinum, followed by gas injection deposition of a platinum layer to protect the milling edge of the lamella and prevent curtaining [52]. The final lamella thickness (~180–280 nm) was achieved by sequentially lowering the ion beam current and distance between milling areas. The overall stability of the lamella was improved by addition of ‘micro-expansion joints’ adjacent to either side of the lamella [53]. A thin 2 nm layer of platinum was applied by sputter-coating to dissipate charge resulting from transmission electron microscopy imaging. Cryotomography of thin lamella was performed on a Titan Krios electron cryomicroscope (Thermo Scientific) operated at 300 keV and equipped with a K3 direct electron detector (Gatan, Inc.) behind a BioQuantum energy filter (Gatan, Inc.) operated with a slit width of 20 eV. Tomograms were acquired using a dose-symmetric tilt scheme [54]. The stage was tilted within the limits of the maximal allowed tilt range for each lamella at 3° tilt steps and a defocus of −4 or −5 μm. The total electron dose per tomogram was 130 electrons/Å^2^. Data acquisition and microscope control was operated using Tomography 5.1 (Thermo Scientific) or SerialEM [55] software. The tilt series raw data were corrected for sample motion using MotionCor2 as implemented in the RELION 3.0 [56,57]. Individual tilt images were aligned and reconstructed using IMOD [28]. Computational alignment was based on the IMOD patch-tracking algorithm and tomography reconstruction was run using the IMOD SIRT-like reconstruction filter setting. Manual image segmentation was performed in IMOD. For their presentation, tomogram z-slices were filtered with a Gauss filter (sigma = 0.6) in Fiji [48].

## Figures and Tables

**Figure 1 ijms-22-13034-f001:**
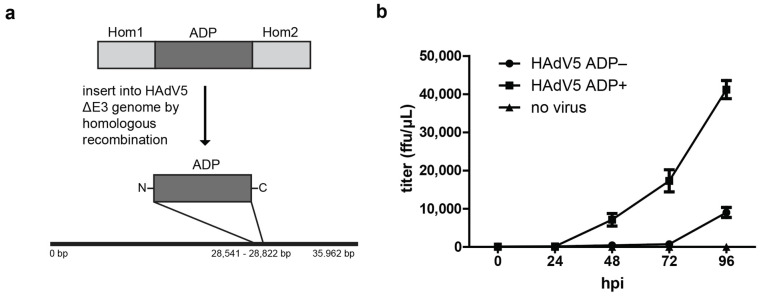
ADP expression accelerates HAdV5 particle release from infected cells (**a**) Overview of ADP insertion into HAdV5 ∆E3 genome by homologous recombination. (**b**) Growth curve of virus particle release from infected A549 cells from 0 hpi to 96 hpi. The two virus strains HAdV5 ADP- and HAdV5 ADP+ and a control in which no virus was used were compared. The supernatant of infected cells was collected at each time-point and their titer was measured as fluorescence-forming units (ffu)/μL by FACS analysis. Three replicates were measured, and the standard deviation is displayed (*n* = 3).

**Figure 2 ijms-22-13034-f002:**
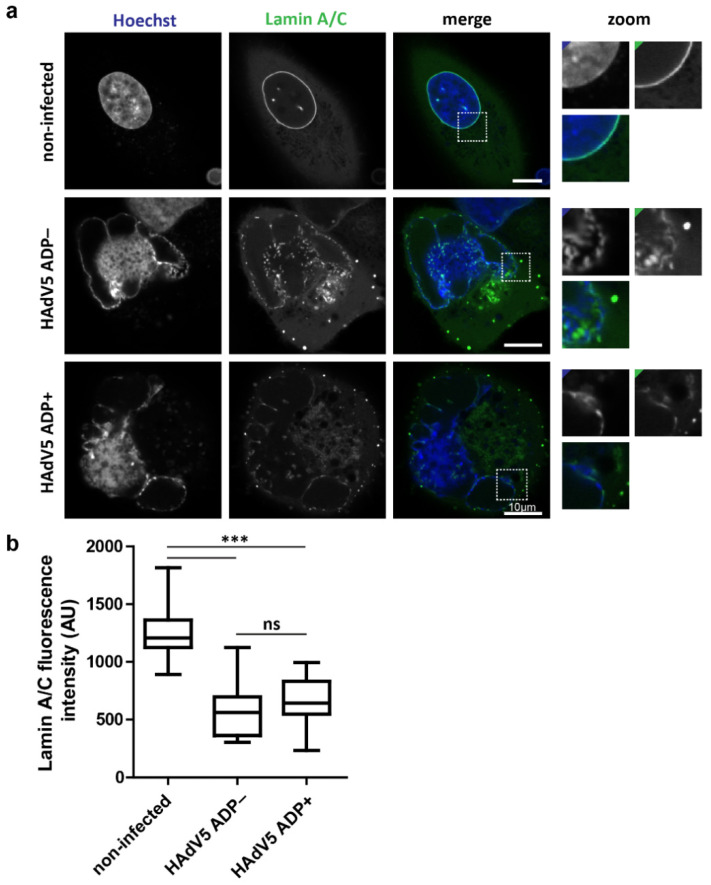
HAdV5 infection reduces lamin A/C signal at the nuclear envelope independently of ADP expression (**a**) A549 cells were infected with HAdV5 ADP− or HAdV5 ADP+ and imaged by live-cell fluorescence microscopy at 48 hpi. The Hoechst 33,342 fluorescence was used to identify infected cells for analysis of their lamin A/C phenotype. A noninfected cell was analyzed in comparison. A representative cell for each phenotype is shown. The dsDNA signal is represented by Hoechst 33,342 stain (Hoechst). The nuclear lamina is represented by an mTagGFP-nanobody recognizing lamin A/C (Lamin A/C). The signal overlap is represented in color (merge). Nuclear membrane regions of interest are enlarged (zoom) with colored corners indicating the channel color. (**b**) Quantification of anti-lamin A/C mTagGFP-nanobody signal intensity at the nuclear envelope in infection with and without ADP. The fluorescence intensity of multiple cells is shown as a boxplot including the median, upper quartile, lower quartile, maximum and minimum of the population (*n* = 20). Statistical significance was calculated using a one-way ANOVA with post hoc Tukey test (ns = not significant; *** = *p* < 0.001).

**Figure 3 ijms-22-13034-f003:**
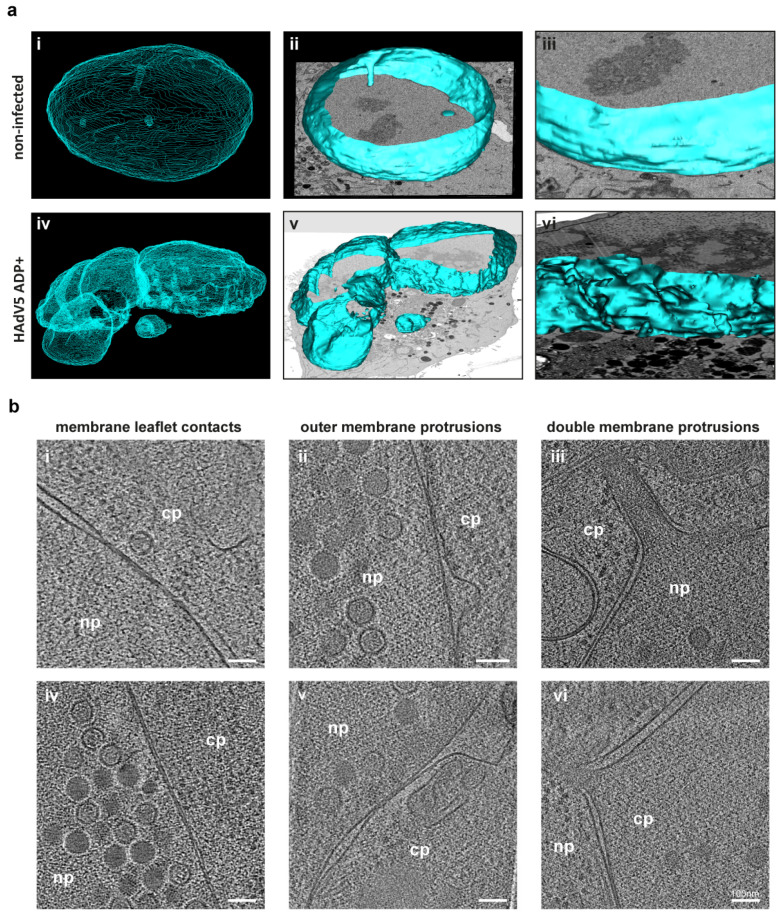
The nuclear envelope in HAdV5 infection loses overall large-scale stability, and electron cryotomography analyses reveal multiple small-scale instability phenotypes. (**a**) A549 cells were infected with HAdV5 pV-mCherry ADP+ [27] and fixed at 42 hpi. The pV-mCherry fluorescence was used to identify infected cells for serial block-face SEM sample preparation. A noninfected cell was analyzed in comparison. The 3D volumes of cells of interest were recorded by serial block-face SEM. The nuclear envelopes of a noninfected and infected cell were manually segmented using the IMOD segmentation tool [28]. A mesh overview of both nuclei is shown in images (i) and (ii). An overlay of a surface rendering onto a SEM z-image is shown in images (iii) and (iv). An enlargement of a surface rendering of the nuclear envelope is shown in images (v) and (vi). (**b**) Electron cryotomograms of multiple membrane modulations observable in HAdV5 pIX-mCherry ADP+ infection. Per phenotype, two example tomograms are shown. Abnormally close contacts between the inner and outer membrane are shown in images (i) and (iv), including a virion detected in the cytoplasm. Outer nuclear membrane protrusions are shown in images (ii) and (v) and double membrane protrusions are shown in images (iii) and (vi). The cytoplasm (cp) and nucleoplasm (np) are labeled.

**Figure 4 ijms-22-13034-f004:**
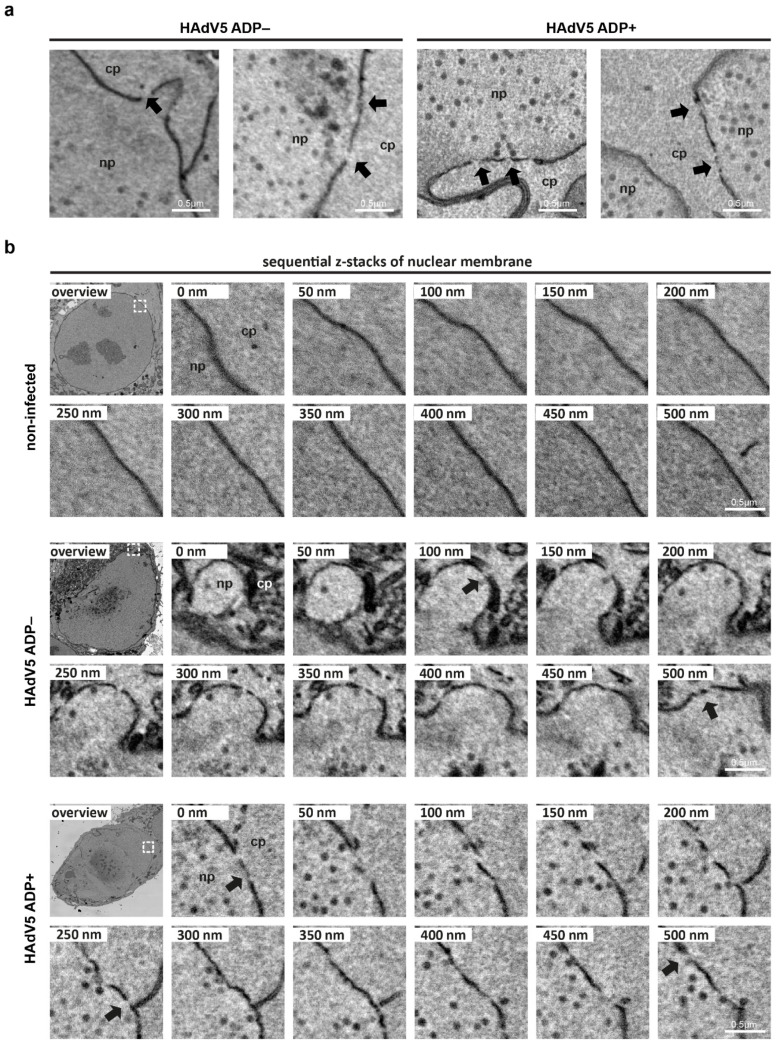
HAdV5 infection induces small membrane lesions detectable by serial block-face SEM. (**a**) Examples of nuclear envelope damage detected in infection with HAdV5 pIX-mCherry ADP− and HAdV5 pIX-mCherry ADP+. A549 cells were infected and prepared for serial block-face SEM at 42 hpi. Arrows indicate sites of damage, and the nucleoplasm (np) and cytoplasm (cp) are labeled. (**b**) Sequence of successive z-slices recorded by serial block-face SEM showing nuclear envelope disruptions by HAdV5 pIX-mCherry ADP− and HAdV5 pIX-mCherry ADP+ infection. A noninfected cell was analyzed for comparison. An overview image for each condition is shown, with a white square indicating a selected ROI. Sequential z-sections of the ROIs are shown, sites of envelope damage are indicated by arrows and the nucleoplasm (np) and cytoplasm (cp) are labeled. A 3D visualization of the membrane damage site of HAdV5 pIX-mCherry ADP+ is shown in Appendix A.

**Figure 5 ijms-22-13034-f005:**
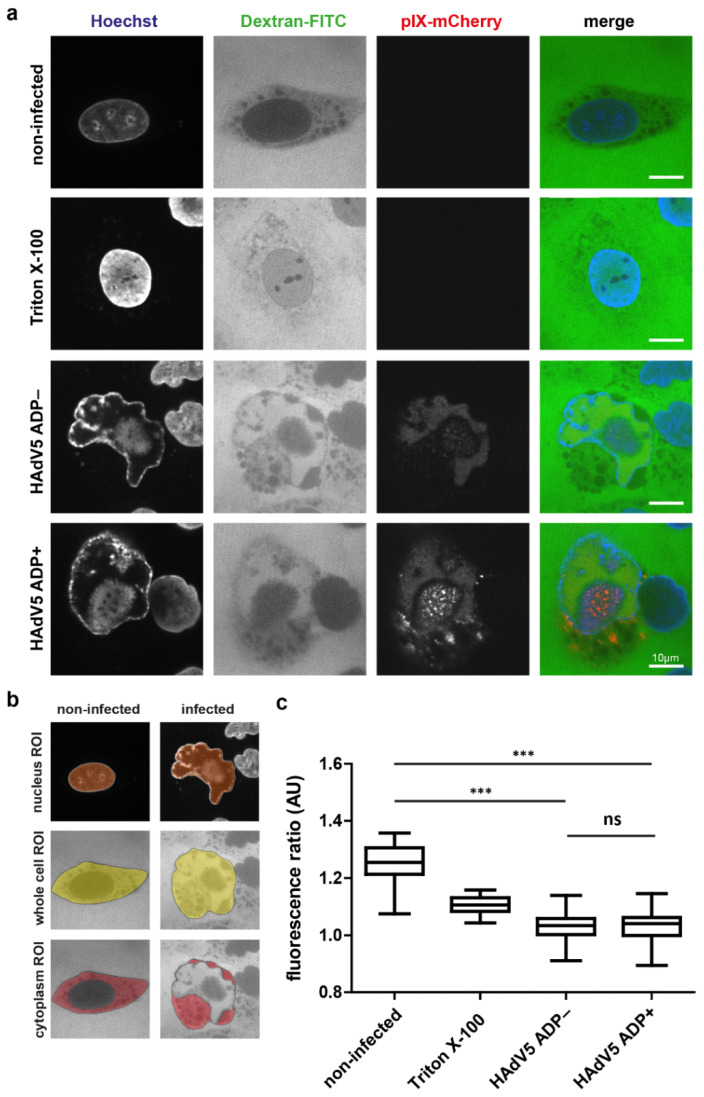
Infection with HAdV5 increases permeability of the nuclear envelope independently of ADP expression. A549 cells were infected with HAdV5 pIX-mCherry ADP− or HAdV5 pIX-mCherry ADP+ and analyzed at 48 hpi. (**a**) Phenotype of A549 cells after incubation in dextran-FITC. The cells were imaged by live-cell confocal spinning-disk fluorescence microscopy. A representative cell is shown for each condition. The dsDNA signal is represented by Hoechst 33,342 stain (Hoechst). The permeability probe dextran-FITC is detected via the FITC fluorescence (Dextran-FITC). Infected cells were detected by positive expression of pIX-mCherry (pIX-mCherry). The signal overlap is represented in color (merge). (**b**) Representation of ROIs for dextran-FITC signal analysis. The nuclear ROI was chosen based on the dsDNA signal, the whole-cell ROI was chosen based on the dextran-FITC cell outline and the cytoplasmic ROI was selected as the whole-cell ROI excluding the nucleus ROI. (**c**) Quantification of dextran-FITC signal ratio between cytoplasm and nucleus of infected cells (*n* = 50). The ratio is shown as a box plot including the median, upper quartile, lower quartile, maximum and minimum of the population. Statistical significance was calculated using a one-way ANOVA with post hoc Tukey test (ns = not significant; *** = *p* < 0.001).

**Figure 6 ijms-22-13034-f006:**
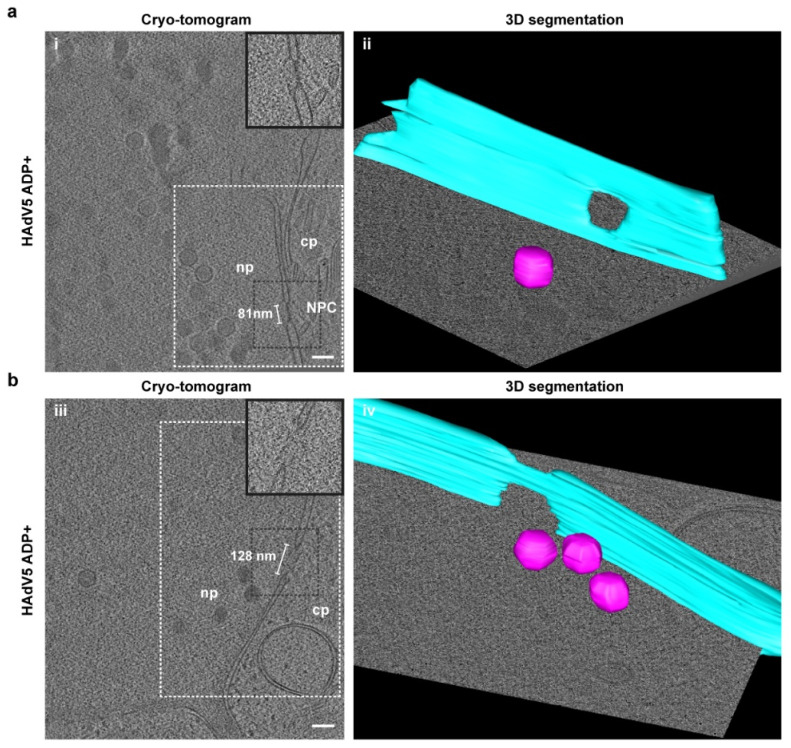
Electron cryotomography reveals nuclear envelope lesion and enlarged nuclear pores in HAdV5 infection. Cryotomograms showing (**a**) a nuclear pore complex (NPC) in an infected cell with a diameter of 81 nm, (**b**) an enlarged nuclear pore with a diameter of 128 nm and (**c**) a site of nuclear envelope disruption with a diameter of 222 nm at its widest opening together with virions outside of the nucleus. For each of the three tomograms, a single z-slice of the reconstructed 3D volume is shown in images (i), (iii) and (v). Areas of nucleoplasm (np), cytoplasm (cp), a nuclear pore complex (NPC) and a lipid droplet (ld) are labeled. The scale bar indicates 100 nm and the measured diameter of the opening for each phenotype is indicated. A zoom-in of the nuclear opening (region indicated by black dotted outline) is shown in the top right corner. A region of interest for each tomogram (region indicated by white dotted outline) is shown in images (ii), (iv) and (vi), respectively. The nuclear envelope (cyan), selected HAdV5 particles (magenta) and a lipid droplet (yellow) were manually segmented using the IMOD segmentation tool [28].

**Figure 7 ijms-22-13034-f007:**
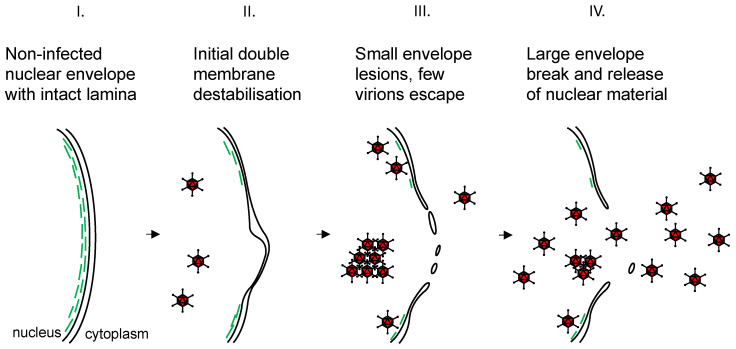
Model of nuclear envelope modulations occurring during HAdV5 nuclear egress. The nuclear double membrane is represented as two separate black lines. (**I**) The noninfected, intact nuclear envelope has a regular, evenly curved appearance. Nuclear lamins (green lines) decorate the inner nuclear membrane and give it stability. (**II**) During infection, the nuclear lamina is degraded and redistributed, likely resulting in a loss of nuclear envelope stability. Consequently, the envelope loses its regular appearance and the distance between the two nuclear membrane layers is altered. (**III**) The nuclear envelope develops small lesions through which few virus particles can escape and can be found in the cytoplasm. Most of the virus particles remain within the nucleus. (**IV**) The small lesions develop into large nuclear membrane breaks. Large quantities of virus and nucleoplasm escape into the cytoplasm (see Appendix A).

**Table 1 ijms-22-13034-t001:** List of primers used for generating ADP virus strains and tagged ADP constructs.

Name	Use	Sequence
ADP ccdB amp fw	PCR	GCTTAGAAAACCCTTAGGGTATTAGGCCAAAGGCGCAGCTACTGTGGGGTTTGCCAGTATACACTCCGCTAG
ADP ccdB amp rv	PCR	GGACAGAAATTTGCTAACTGATTTTAAGTAAGTGATGCTTTATTATTTTTTTTTACAGCCCCATACGATATAAGTTG
ADP rescue fw	PCR	GCTTAGAAAACCCTTAGGGTATTAGGCCAAAGGCGCAGCTACTGTGGGGTTTATGACCAACACAACCAACG
ADP rescue rv	PCR	GGACAGAAATTTGCTAACTGATTTTAAGTAAGTGATGCTTTATTATTTTTTTTTATCATACTGTAAGAGAAAAGAACATGTG
ADP-HA ccdB amp fw	PCR	GAATCCATAGATTGGACGGACTGAAACACATGTTCTTTTCTCTTACAGTAGCCAGTATACACTCCGCTAG
ADP-HA ccdB amp rv	PCR	ATTTGCTAACTGATTTTAAGTAAGTGATGCTTTATTATTTTTTTTTATCACAGCCCCATACGATATAAGTTG
ADP-HA rescue	Direct Red Rec.	GAATCCATAGATTGGACGGACTGAAACACATGTTCTTTTCTCTTACAGTATACCCATACGATGTTCCAGATTACGCTTGATAAAAAAAAATAATAAAGCATCACTTACTTAAAATCAGTTAGCAAATTTCTG

## Data Availability

All data relevant to this work are provided in the manuscript as much as possible. Full (un)processed cryoelectron tomograms and other raw imaging materials are available upon request.

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
