# Peer review of "Human Adenovirus Type 5 Infection Leads to Nuclear Envelope Destabilization and Membrane Permeability Independently of Adenovirus Death Protein"

_ijms, 2021, doi:10.3390/ijms222313034_

Round 1

Reviewer 1 Report

In their article “Human adenovirus type 5 infection leads to nuclear envelope destabilization and membrane permeability independently of adenovirus death protein”, Pfitzner et al. examine the impact of the adenovirus death protein (ADP) on nuclear memebrane integrity during Ad5 infection. The stability of the nuclear membrane was analyzed in the presence and absence of ADP using different microscopic methods. The manuscript is clearly structured and well written. The experiments are conducted properly, and results are presented in a clear and comprehensible manner. However, results presented in this study are solely descriptive, and mechanistic data are missing. While the manuscript is suitable for publication, I encourage the authors to address the following points to strengthen their manuscript.

  1. The authors addressed the localization of ADP in an infected cell. While for Ad2 ADP a nuclear localization has been described, no such localization could be obtained for C-terminally tagged Ad5 ADP. As discussed, the position of a tag needs to be carefully evaluated since it might impact both, the correct localization and function of the respective protein. Therefore, I am wondering why the authors did not compare N-terminal and C-terminal tagging. Further, considering localization artefacts of ADP due to tagging, would Ab-staining of the native ADP have been an option of localization studies?
  2. Throughout the manuscript, the authors repeatedly compare their obtained data with those availabve for Ad2 ADP. To what extent differ Ad5 and Ad2 ADP in their three-dimensional structure and linear sequence (localization signals such as NLS)?
  3. Paragraph 2.2/ Figure 2A. Please include a more detailed description of the observed phenotypes in the text, as it is challenging to see the differences, especially between phenotypes A and B. Also, the authors state a co-localization with th ER (page 5, line 149). What marker was used to microscopically identify the ER?
  4. Paragraph 2.3. The authors observe a reduction of lamin A/C expression in infected cells 48 hpi that occurred independently of ADP. Since lamins are crucial for nuclear membrane stability, it would have been interesting to monitor the lamin levels throughout an infection cycle. Lamins can be phosphorylated at different sites, and phosphorylation patterns determine their structural properties. Therefore, it would have been informative to analyze lamin A/C phosphorylation in the presence or absence of certain Ad proteins (such as ADP) or Ad5 in general.
  5. Page 9, line 269. It should be “images iii and vi”.

Author Response

Thank you for your valuable comments. Please check my reply in the attachment. Thanks.

Reviewer 2 Report

In the manuscript submitted by Pfitzner et al., the authors aimed to elucidate the mechanisms of HAdV5 release from infected cells, in particular the role of the viral protein ADP.

From their experiment shown in Fig.1b it is clear, that ADP is an essential factor to increase viral release from infected cells. The additional in silico predictions shown in Fig.1 are OK, but remain speculative without biochemical proof. Could be also moved to supplement.

Next the authors set out to study the localization of tagged ADP in transfected H1299 cells, and found divergent localization, both between mCherry and HA tag, as well divergent from previous literature.

I would also ask to explain why this experiment is done in H1299 and not A549 cells.

Their findings are not too surprising, as high overexpression of transmembrane proteins often lead either to aggregation in the ER/Golgi (as seen in Fig. 2a ("phenotypes A and B") and/or misfolded protein (probably "phenotype C"). Especially as the ADP is only 10kD as compared to ~30kD of mCherry. What is described as "perinuclear vesicles" looks like Golgi apparatus to me. A co-staining with Golgi-and ER markers would clarify that.

It is also not possible to discriminate in Phenotype "A" whether it is outer or inner nuclear membrane - which makes a huge difference regarding functionality. The mCherry tag may block the nuclear import of ADP by steric hinderance. As a control I would suggest tagging ADP on the N-terminus and compare.

Unfortunately Fig. 2d, showing the HA-tagged construct now again in A549 cells (why not in H1299?) is hard to interpret as only a single cell is shown. Do all cells/nuclei look like that?  I also do not agree with the statement, that ADP-HA is not seen at the nuclear envelope, as there are punctate signals around the "nucleus". Again, comparison to an N-terminal HA-tagged ADP would be important.

Finally, a Western blot proofing correct size of the contructs is missing and must be provided.

Overall, the presented data is not conclusive, as localization of ADP remains unclear and what has been seen by the authors might be overexpression or tagging artefacts.

In the next experiment the authors study the impact of HAdV5 infection on nuclear envelope morphology - although to be correct, it is only the nuclear lamina which is visualized by lamin A/C. I am not sure, if really the total amount of lamin A/C is reduced, as the nuclei are blown up due the infection, leading to an increased surface and thereby diluting lamin A/C -> which is measured as reduced signal intensity. I would suggest, that the authors measure total intensity of one nucleus and compare. Even better would be a Western Blot showing lamin A/C amounts in control and infected cells. A comparison with lamin B and a true nuclear envelope protein - e.g. Emerin or Lap2beta would be interesting as well.

Overall, there is no obvious difference in nuclear morphology, if ADP is present or not.

Line 244: ..., the the ADP+... (delete one "the")

In Figures 4 and 5, cell infected with HAdV5 pIX-mCherry-ADP were processed for cryo tomography. Although the images are of high quality, it remains the strong doubt that the mCherry-ADP is not functional, hence no difference between + and -ADP would be observed.

Basically the same is true for the data presented in Fig. 6.

An essential control that is missing in this respect, is the viral yield as done in Fig. 1b -> wt HAdV5 compared to HAdV5 +mCherry-ADP -/+

I sorry that I have to recommend rejection of the manuscript in its current state, solely because most assumptions are based on an ADP-construct which might be non-functional - at least it was not proven, that it is functional.

The assays and methods are generally of high quality and nuclear envelope perturbation by HAdV5 infection might be worth publishing on their own.

The role of ADP in this respect would be highly interesting and a clear plus for the manuscript, but the used constructs have to be vigorously characterized and tested before building further experiments on top.

Author Response

(The authors gave the same response as above.)

Round 2

Reviewer 2 Report

The authors have put significant efforts into improving the manuscript and clarifying the data description and interpretation, and meets the criteria for publication in my opinion.

I also apologize for my misunderstanding of the different constructs in the first review round!